# Midterm Results of Iliac Branch Devices in a Newly Established Aortic Center

**DOI:** 10.3390/life12081154

**Published:** 2022-07-29

**Authors:** Sarolta Borzsák, András Süvegh, András Szentiványi, Daniele Mariastefano Fontanini, Milán Vecsey-Nagy, Péter Banga, Péter Sótonyi, Zoltán Szeberin, Csaba Csobay-Novák

**Affiliations:** 1Department of Interventional Radiology, Semmelweis University, 1122 Budapest, Hungary; borzsak.sarolta@med.semmelweis-univ.hu (S.B.); suviandris@gmail.com (A.S.); sz.andris1@gmail.com (A.S.); fontanini.daniele@med.semmelweis-univ.hu (D.M.F.); vecsey_nagy.milan@med.semmelweis-univ.hu (M.V.-N.); 2Semmelweis Aortic Center, Heart and Vascular Center, Semmelweis University, 1122 Budapest, Hungary; banga.peter@med.semmelweis-univ.hu (P.B.); sotonyi.peter@med.semmelweis-univ.hu (P.S.); szeberin.zoltan@med.semmelweis-univ.hu (Z.S.); 3Department of Vascular and Endovascular Surgery, Semmelweis University, 1122 Budapest, Hungary

**Keywords:** iliac aneurysm, endovascular procedures, iliac branch device

## Abstract

The first-line treatment of common iliac artery aneurysms is endovascular repair. International guidelines recommend the preservation of the internal iliac artery, which is best achieved by the implantation of an iliac bifurcation device (IBD). Our aim was to evaluate the initial midterm results of IBDs in the leading vascular center of Hungary. In this single-center retrospective study, relevant clinical data and the results of the imaging examinations were collected and analyzed in all patients who underwent IBD implantation between December 2010 and July 2021. Thirty-five patients (31 males, mean age: 67.9 ± 8.5 years) underwent endovascular treatment with 37 IBD implantations. Technical success was achieved in 88.2% of the patients, with no perioperative mortality or open surgical conversion. One patient was lost during follow-up. Internal iliac artery occlusion was detected in three (8.8%) patients, and reintervention was performed in five (14.7%) patients. Primary patency of the internal iliac branch was 97.1% at 1 month, 93% at 2 months, and 89.0% at 5 years. The average follow-up time was 20.1 ± 26.2 months, during which two (5.9%) deaths occurred. Our initial experience with iliac branch devices was associated with a low complication rate and a favorable outcome, which confirms the midterm success of this intervention.

## 1. Introduction

As endovascular treatment possibilities evolve, the management of aortic and aorto-iliac pathologies is shifting towards endovascular procedures in patients with suitable anatomy [1]. On the other hand, extensive iliac aneurysm repair might not provide a durable exclusion of the aneurysm, or it might endanger pelvic circulation [2]. Recent guidelines recommend the preservation of at least one internal iliac artery to minimize the risk of ischemic complications following the loss of the internal iliac arteries. In addition to a surgical approach, various endovascular techniques can be used to preserve hypogastric anatomy, e.g. the bell bottom technique, sandwich technique, and multiple side branch techniques. However, it can be best obtained by the implantation of an iliac branch device (IBD) [1,3,4]. Several studies have reported on the outcomes of IBDs, demonstrating favorable results [2,5,6]. However, the availability of such devices shows significant geographical differences due to the lack of reimbursement and/or centralization, especially in Eastern European countries [7]. Therefore, as such data are currently missing from the literature, we aimed to evaluate the initial experience of IBD implantations regarding the short- and midterm results at a pioneer aortic center in Hungary. 

Our aim was to examine the results of these interventions, above all the per vessel technical success rate, technical success rate, and clinical success rate, and to describe the outcome parameters at follow-up, such as aortic-related and all-cause mortality, need for reintervention, and patency of the iliac arteries.

## 2. Materials and Methods

### 2.1. Study Population

We performed a retrospective analysis of all consecutive patients who underwent IBD implantation between December 2010 and July 2021. The study was approved by the local ethics committee (Semmelweis University Regional and Institutional Committee of Science and Research Ethics: 92/2021) and performed in accordance with the Declaration of Helsinki. All patients provided informed consent.

Demographic data and cardiovascular risk factors, as well as anatomical, procedural, and postoperative variables, were collected retrospectively. Follow-up clinical examinations and imaging were performed according to current guidelines: first at 30 days, then at 6 months, and then yearly depending on the results of the computed tomography angiography (CTA) examination completed during the first follow-up. In patients with severely impaired kidney function, magnetic resonance angiography (MRA) was performed instead of a CTA.

### 2.2. IBD Procedure

The IBD deployment was performed as an adjunctive procedure during an endovascular aneurysm repair (EVAR) if aorto-iliac involvement was seen, or as a stand-alone procedure, when only an isolated iliac artery aneurysm was repaired. The choice of implanted bifurcation device was based on the patients’ + anatomic features and the availability of the different IBDs. Planning was performed using IntelliSpace Portal (Philips Healthcare, Best, The Netherlands) or 3Mensio Vascular software (Pie Medical Imaging B.V., Maastricht, The Netherlands). Zenith Branch Endovascular Graft (Cook Medical, Bloomington, IN, USA), Gore Iliac Branch Endoprosthesis (IBE; W.L. Gore & Associates, Newark, DE, USA), and Jotec E-liac (Jotec GmbH, Hechingen, Germany) were used. The Cook device was preferred for smaller common iliac luminal diameters, whereas Gore implants were used for wider lumina. Jotec devices were preferred when isolated repair was planned, and proximal diameters were suitable. 

All procedures were performed by two physicians (CCN, ZSz), both of whom are proctors of a firm. A fixed X-ray imaging system was used, and latter cases were performed in a hybrid operating room. Open surgical cutdown was preferred in our early experience, with a shift towards the percutaneous technique using Perclose Proglide (Abbott Laboratories, Abbott Park, IL, USA) suture-mediated closure system. Additional collagen-plug based vascular closure devices (AngioSeal VIP; Terumo Corporation, Tokyo, Japan) were used liberally if suture-mediated vascular closure failed. General or locoregional anesthesia was used at the discretion of the anesthetist. Postoperative course was usually managed outside the intensive care unit. Dual antiplatelet therapy was maintained postoperatively for three months followed by lifelong aspirin or clopidogrel monotherapy.

### 2.3. Data Analysis

In terms of terminology, measurement techniques, and outcome parameters, we followed definitions within the most recent reporting standards document published by Oderich et al. Technical success was considered to be achieved if successful access to the arterial system was obtained, the stent graft components were deployed, and the preservation of all branches was successful, and no type I or III endoleak was seen on the 30-day follow-up imaging study. A clinical success was defined as the absence of important disabling permanent clinical sequelae, such as aortic-related complications or permanent paraplegia, disabling stroke, or permanent dialysis in addition to technical success [8]. Primary endpoints in this study were aortic-related and all-cause mortality, need for reintervention, and patency of the iliac arteries. Secondary outcomes were technical and clinical success, detection of endoleaks, and major adverse events, including new-onset renal failure, major stroke, myocardial infarction, respiratory failure, and significant buttock claudication.

### 2.4. Statistical Analysis

Categorical variables are presented as numbers and percentages, and continuous parameters are reported as mean ± standard deviation (SD). Kaplan–Meier survival estimates were calculated to assess long-term outcomes (patency, re-intervention, and survival); the curve is displayed up to a value of standard error (SE) < 0.10. A value of *p* < 0.05 was considered statistically significant for all measurements. Statistical analyses were carried out using IBM SPSS (Armonk, NY, USA, version 27.0) and GraphPad Prism 8 (GraphPad Software, San Diego, CA, USA), and the latter was used to graph data.

## 3. Results

Between 14 December 2010 and 23 July 2021, 37 IBDs were implanted in 35 patients in a tertiary care university medical center. The primary disease was aorto-iliac aneurysm in 19 cases, isolated iliac aneurysm in 11, chronic aortic dissection in 3 and Ib endoleak following an EVAR in 2 cases. In the 11 cases where the indication of the IBD deployment was an isolated common iliac aneurysm, a stand-alone IBD implantation was performed. The remaining 24 patients were treated in conjunction with an EVAR. Three patients also underwent a thoracic endovascular aortic repair (TEVAR) procedure for a thoracic aortic aneurysm in addition to the EVAR-IBD implantation. The mean age was 67.9 ± 8.5 years, and patients were mostly male (89%). The population and aneurysm characteristics of patients undergoing IBD implantations are reported in Table 1. Detailed procedural data are shown in Table 2.

Twenty patients (57.1%) were treated outside of the instructions for use (IFU). Based on the IFU, only the Jotec E-iliac graft should be used in isolated iliac aneurysms; however, in six cases, a Cook ZBIS or a Gore IBE endograft was placed and isolated, due to proximal landing zone diameter issues. The other 14 patients were outside of the IFU, either because of aortic dissection as their primary disease or because they did not meet the anatomical requirements of the IFUs. In these cases, an aortic team decision was made to recommend IBD implantation, to which the patient consented. Off-label/non-IFU repairs were equally prevalent throughout the study period.

Our per vessel technical success rate was 100%, and none of the internal iliac arteries were lost. The overall technical success rate was 88.2%. The primary clinical success rate was 82.4%, while the assisted primary clinical success rate was 88.2%.

The mean postoperative hospitalization duration was 4.6 ± 0.7 days, and the average length of the intensive care unit stay was 0.3 ± 0.5 days. The mean follow-up time was 20.1 ± 26.2 months. One patient was lost during follow-up. During the follow-up period, no peri-operative or in-hospital deaths were recorded, nor was surgical conversion needed. There was no myocardial infarction, stroke, new-onset renal failure, mesenteric or spinal cord infarction, respiratory failure, or significant buttock claudication.

Freedom from IBD occlusion values were 97.1%, 93.5%, and 89.0% at 1, 2, and 4 months using Kaplan–Meier estimates, respectively (Figure 1). In total, three iliac occlusions were observed, and only the internal iliac branch was affected. All the occlusions were left untreated. 

Seventeen endoleaks were detected in 14 patients. One type I, one type V, and two type III endoleaks were found, while 10 of the patients had a type II endoleak. Five re-interventions were necessary (14.7%). Endoleaks were managed when a significant aneurysm sac growth (>5 mm) was seen (4 cases, 11.4%). In three cases, successful embolization was performed (using histoacryl and lipiodol), but in one case, the source of the endoleak could not be clearly identified. The need for re-intervention was related to the IBD device in four patients (11.8%).

Two late deaths were recorded, neither of them related to the endovascular intervention or the aneurysm. The cause of death was gastro-intestinal bleeding in one case and Clostridium sepsis in the other case, both of which occurred months after the IBD procedure. The freedom from all-cause mortality and freedom from aneurysm-related mortality was 92.4% and 100%, respectively (Figure 2).

## 4. Discussion

Preserving the internal iliac artery during EVAR or during an isolated iliac aneurysm treatment is advocated to minimize the risk of pelvic ischemic complications [9]. IBDs have been used as an adjunctive procedure during an EVAR and as a stand-alone procedure for over a decade with excellent results [6,10].

In recent years, the numbers of IBDs started to rapidly increase due to the establishment of a multidisciplinary aortic center. In Figure 3, we provide a graph demonstrating the number of IBD implantations performed at our institution each year. Despite the lack of formal centralization in Hungary regarding both standard and complex aortic procedures, our institute is a pioneer in the aortic field. We have performed 90% of the complex aortic procedures for more than 80% of IBD cases in Hungary so far [11].

The results of our initial series of patients are favorably compared with other reported data from experienced aortic centers in Western Europe. The technical success rate was 88.2% in our study. In their systematic review, Kouvelos et al. reported a technical success rate of endovascular internal iliac artery preservation in 96.2% of cases [9]. In a study by Simonte et al., including 149 patients with 157 IBD implantations and a median follow-up of 34.0 months, the technical success rate was 97.5% [6]. Parlani et al. reported a technical success rate of 95% [2]. Haulon et al. achieved a technical success rate of 94% [4]. Mylonas et al. demonstrated an outstanding technical success rate of 100%, although they reported their results in accordance with more permissive criteria [12].

However, the existence of a learning curve is a well-known fact regarding all procedures, which explains our slightly inferior outcome parameters. Simonte et al. performed a sub-analysis comparing outcomes achieved in the first 25 IBD deployments, and those observed in the later phase. Significant differences were detected—the peri-operative success rate was 84.0% in the first period, and it was 97.7% after the first 25 cases [6]. The study of a 5-year experience on IBD implantations conducted by Parlani et al. also confirmed the important role of the learning curve effect, as they detected four out of the five technical failures during their first year of experience with IBDs [2]. Compared to their five intra-operative IBD internal limb occlusions, our per vessel technical success rate of 100% shows a better technical outcome.

Another factor that might explain the slightly inferior outcome rates of the devices is the high number of patients treated outside the IFU (57.1%). Off-label use was most commonly associated with a reduced diameter of the common iliac bifurcation. In particular, we believe the 16 mm threshold for the Cook ZBIS device is rather strict, and narrow iliac bifurcations down to 12–13 mm may be treated successfully with an acceptable outcome. These procedures are technically more demanding, but outcomes may not be inferior to on-label cases once the technical difficulties are managed intraoperatively and proper post-dilation is performed, most commonly with a kissing balloon maneuver. Similarly, narrow aortic bifurcation was found to be non-inferior regarding long-term outcome if a proper implantation technique was used [13]. 

There is an interesting study by Tomczak et al. that aimed to evaluate the number of patients with asymptomatic abdominal aortic aneurysms, regardless of the treatment plan, who can be treated by EVAR with stentgraft devices commercially available in East–Central Europe in conformity with the IFU. The suitability rates of the examined devices varied from 20% to 65%. It was found that 32% of the patients were not suitable for any of the analyzed stentgrafts, assuming a rigorously followed IFU [14]. Similar difficulties could be present regarding the armamentarium of IBDs, limiting the patients who can be endovascularly treated within the IFU. The liberalization of morphology indications might result in increased failure rates and higher endoleak rates [2]. In a comparative study by Donas et al., where minimal anatomical characteristics were used for IBD implantation and challenging anatomies of the internal iliac artery were also included, a higher endoleak rate was observed (12.5%) than the average literature data [15]. 

On the other hand, Simonte et al. found similar results when comparing the long-term outcomes of IBD implantations performed in an experienced center as per or outside manufacturer’s IFU [16]. Rodriguez et al. reported similar findings: in a study where 15 patients were treated within the IFU and 24 patients’ IBD implantations were non-IFU, no significant difference was found regarding technical success and device-related reintervention in the short term [17]. Another approach, when patients with challenging anatomy require iliac aneurysm treatment, could be the use of a custom-made iliac branch device. Huang et al. found non-inferior results when comparing their custom-made devices to commercial devices in a cohort of 46 patients [18].

Our internal iliac artery occlusion rate of 8.8% at 2–5 years is comparable to a few other studies. Haulon et al. and Karthikesalingam et al. both reported similar, slightly elevated occlusion rates of 11.3–12.2% [4,19]. However, our iliac patency rate was lower than what was mostly found in other similar studies, where the internal iliac branch patency was between 89.7% and 100% [12,20].

Our endoleak rate with 17 detected endoleaks was higher than the literature data. We detected 10 type II endoleaks in our patient cohort of 35 compared with the results of the D’Oria et al. study on the bilateral use of IBDs within the pELVIS registry, where only 17 persistent type II endoleaks were seen in 96 patients [21]. However, the number of endoleaks, which required invasive therapy, did not differ much from the existing data. We only treated endoleaks with a significant aneurysm sac growth, which was the case in four patients. We find it important to try to manage complications conservatively, especially in fragile patients. One possibility is to modify the patient’s medication; e.g., we had a case, in which a type I endoleak disappeared after the dual antiplatelet therapy was changed to a mono antiplatelet therapy.

The re-intervention rate of 14.7% is also comparable to the existing data in the literature. Verzini et al. reported a re-intervention rate of 18.2% [22]. Gibello et al. found a re-intervention rate of 11.8% in patients with a common iliac artery diameter <18 mm and 19.1% in those with a common iliac artery diameter ≥18 mm [23]. Overall, 42 re-interventions were performed among the 575 patients (7.3%) in the patient cohort analyzed by Donas et al.

Most authors agree that the outcome of an open abdominal aortic aneurysm repair is associated with surgeon and hospital caseload [23,24,25,26,27,28]. McPhee et al. found that after an elective open abdominal aortic aneurysm repair, surgeon case volume is the primary determinant of in-hospital mortality [24]. An international analysis of 178,860 patients found no volume effect on in-hospital or 30-day mortality after EVAR for abdominal aortic aneurysm [26]. Mortality after EVAR was unaffected by either surgeon or hospital volume in the Australian population studied by Sawang et al., but hospital volume in the TEVAR subgroup showed a strong inverse correlation with mortality [25]. Complication rates and in-hospital mortality following abdominal aortic aneurysm repairs were found to be inversely associated with annual hospital volume in Germany [29]. After EVAR, hospital volume was associated with slightly higher perioperative mortality in the study of Zettervall et al., but no such association was observed for surgeon volume [26]. 

A recent Dutch analysis also showed a significant effect of hospital volume on perioperative mortality following complex EVAR, with high volume centers demonstrating decreased mortality rates [30]. D’Oria et al. investigated the association between hospital volume and failure to rescue after EVAR and open aortic repair of intact abdominal aortic aneurysms and found a significant association: hospitals in the top volume quartiles achieve the lowest mortality after a complication has occurred [31]. 

To our best knowledge, no data on the effect of surgeon case volume or hospital volume are available regarding the outcomes of IBD implantations. However, our cases being analyzed by only two physicians, both of whom are proctors, and our results being slightly better than other centers’ initial data, suggest that the operator’s experience (both prior endovascular experience and practice obtained during the IBD implantations) might have an effect on decreasing the learning curve.

### Study Limitations

We acknowledge the limitations of our study. Our single-center, retrospective analysis includes a relatively small sample size of patients, and since the vast majority of these IBDs were deployed in the past three years, and the COVID-19 pandemic delayed many control examinations, we have a significant number of patients with short follow-up data; the follow up completion rate has been relatively low recently. Patient and material selection for intervention were derived from team discussions; we did not have a standardized approach. The heterogeneity of the patients regarding the type of treated pathology also limits the generalizability of our results. 

Furthermore, three different manufacturer’s endograft models were utilized in our study. It is possible that differences in peri-operative or late performances among the grafts may exist, but we did not have enough data in this study to perform subgroup analyses. Nonetheless, to our best knowledge, no relevant differences were detected among the current IBDs regarding patient outcomes [12,18]. Finally, the low event rate did not make the evaluation of the adjusted risk factors for the primary and secondary endpoints possible.

## 5. Conclusions

In this retrospective study, a high technical success rate and low complication rate were found with a high freedom from disease-related mortality when analyzing our short- and midterm results, despite observing the initial cases of our center. The safe introduction of IBDs for the treatment of iliac aneurysms could be the result of the few physicians performing the implantations and their previous expertise in endovascular procedures.

## Figures and Tables

**Figure 1 life-12-01154-f001:**
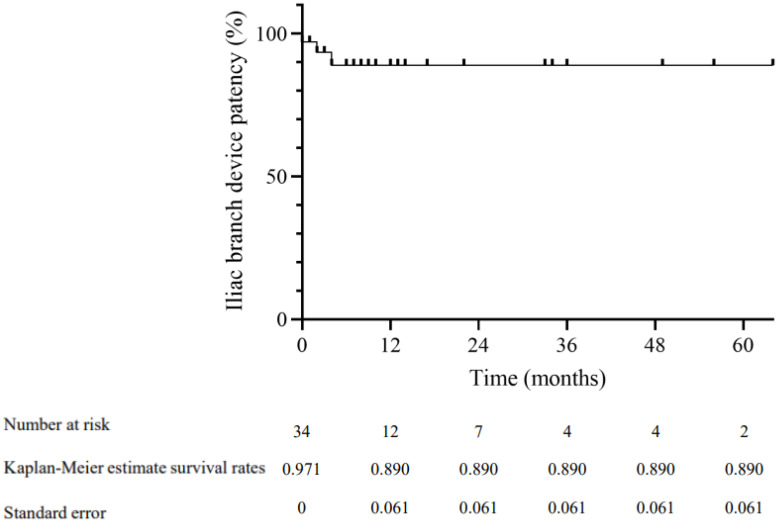
Kaplan-Meier estimates of iliac branch patency treated by iliac branch devices.

**Figure 2 life-12-01154-f002:**
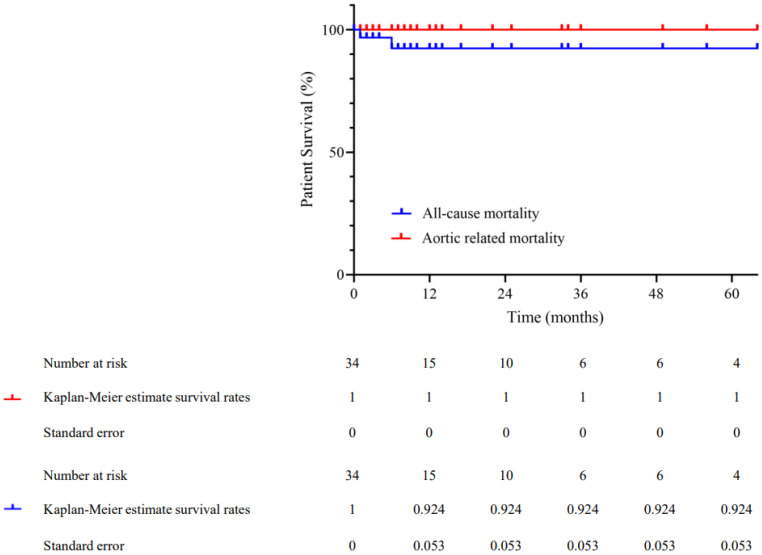
Kaplan-Meier estimates of all-cause mortality and aortic related mortality treated by iliac branch devices.

**Figure 3 life-12-01154-f003:**
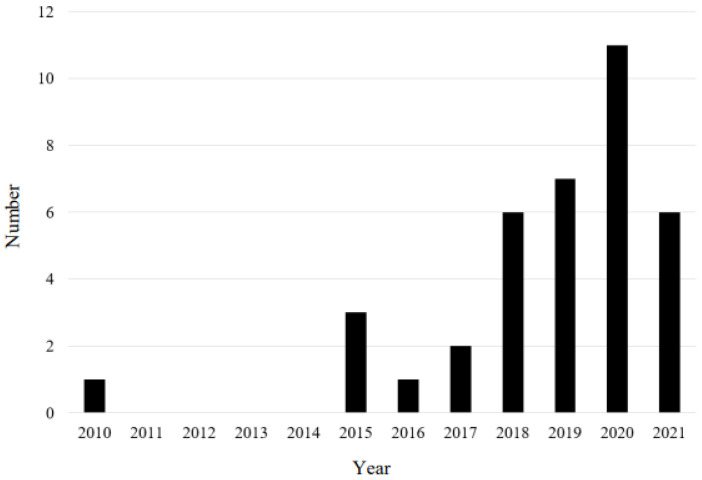
IBD implantations in our institution.

**Table 1 life-12-01154-t001:** Baseline patient and anatomical characteristics.

Variable		N (%) or Mean ± SD
Demographics	Male gender	31 (89)
	Mean age, years	67.9 ± 8.5
	BMI, kg/m^2^	28.5 ± 5.7
Cardiovascular risk factors	Hypertension	35 (100)
	Current smoking	13 (37)
	Hypercholesterolemia	16 (46)
	Diabetes mellitus	6 (17)
	Peripheral artery disease Cardiac disease	7 (20) 18 (51)
	Chronic obstructive pulmonary disease	10 (29)
	Chronic kidney disease stage III-V	11 (31)
	Previous aortic repair	12 (34)
	Prior malignancies	11 (31)
Anatomical characteristics	Left CIA aneurysm diameter, mm	32.3 ± 14.1
	Right CIA aneurysm diameter, mm	35.0 ± 13.5

Abbreviations: N = number; SD = standard deviation; BMI = body mass index; CIA = common iliac artery.

**Table 2 life-12-01154-t002:** Baseline procedural characteristics.

Variable		N (%) or Mean ± SD
Implanted devices	Cook ZBIS	20 (54)
	Gore IBE	12 (32)
	Jotec E-iliac	5 (14)
	Isolated IBD	11 (31)
	Bilateral IBD	2 (6)
Procedural data	Contrast dose, mL	139.25 ± 71.36
	Fluoroscopy time, s	2832.55 ± 1656.08
	Dose area product, Gy*cm^2^	294.45 ± 442.74
	Total length of hospital stay, days	4.60 ± 0.69
	Length of intensive care unit stay, days	0.3 ± 0.51
Complications	Type I endoleak	1 (3)
	Type II endoleak	10 (29)
	Type III endoleak	2 (6)
	Type IV endoleak	0 (0)
	Type V endoleak	1 (3)

Abbreviations: N = number; SD = standard deviation; IBD = iliac branch device.

## Data Availability

Not applicable.

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
