# Peer review of "Midterm Results of Iliac Branch Devices in a Newly Established Aortic Center"

_life, 2022, doi:10.3390/life12081154_

Round 1
Reviewer 1 Report
Interesting study, despite lo sample size it might be suitable for the journal.
1. How many surgeons performed procedures?
2. Antiplatelet/anticoagulant treatment after the procedure? Some patients might have indications due to atrial fibrillation or PCI
3. Significantly less optimistic outcome is presented in abdominal aortic aneurysm, it might be important do discuss this topic and compare outcomes https://pubmed.ncbi.nlm.nih.gov/33868423/
Reviewer 2 Report
Dear authors,
Original paper, congratulations! I would like you to add some paragraphs about the Endoleaks, as you found 17, almost 50% of your patients population.
Reviewer 3 Report
Authors tested the midterm results of iliac branch devices in an aortic center and proved their results are with high success. Some of the content needs to be modified to make the paper with higher quality.
1. Line 22: the writing contains several errors in the sentences “The average follow-up time was 20.1 ± 26.2 months. during which 2 (5.9%) deaths occurred Our initial experience…”. “during which…” is a new sentence, the first character should be capital. “Our initial experience…”, no full stop is before this sentence. Please double check the writing in this line for the abstract.
2. Line 109: please use the standard writing in manuscript: SE<0.10, p<0.05.
3. Line 115: please modify the date format like 14th, December 2020 and 23rd, July 2021.
4. Line 152: when citing figures, please use standard writing way: (Figure 1 or Fig. 1), no dot after the number. The same for other figures’ citation in the text.
5. Figure 1: if the bottom part is a table, please use the three-line format for table, demonstrate the table as an independent one.
6. When cite references in the main text, please note that no dot should be in front of the reference number. Like line 211, 215. Keep the format consistent in the manuscript.
Round 2
Reviewer 2 Report
For me it’s ok